# Analyzing the Behavior and Financial Status of Soccer Fans from a Mobile Phone Network Perspective: Euro 2016, a Case Study

**Gergő Pintér** * and **Imre Felde**

John von Neumann Faculty of Informatics, Óbuda University, Bécsi út 96/B, 1034 Budapest, Hungary;
felde.imre@uni-obuda.hu

* Correspondence: pinter.gergo@uni-obuda.hu

**Abstract:** In this study, Call Detail Records (CDRs) covering Budapest for the month of June in 2016 were analyzed. During this observation period, the 2016 UEFA European Football Championship took place, which significantly affected the habit of the residents despite the fact that not a single match was played in the city. We evaluated the fans' behavior in Budapest during and after the Hungarian matches and found that the mobile phone network activity reflected the football fans' behavior, demonstrating the potential of the use of mobile phone network data in a social sensing system. The Call Detail Records were enriched with mobile phone properties and used to analyze the subscribers' devices. Applying the device information (Type Allocation Code) obtained from the activity records, the Subscriber Identity Modules (SIM), which do not operate in cell phones, were omitted from mobility analyses, allowing us to focus on the behavior of people. Mobile phone price was proposed and evaluated as a socioeconomic indicator and the correlation between the phone price and the mobility customs was found. We also found that, besides the cell phone price, the subscriber age and subscription type also had effects on users' mobility. On the other hand, these factors did not seem to affect their interest in football.

**Keywords:** mobile phone data; call detail records; type allocation code; data analysis; human mobility; large social event; social sensing; socioeconomic status

## 1. Introduction

Football is one of the most popular sports worldwide and the European and World Championships, especially the finals, are among the most watched sporting events around the globe. The Euro 2016 final was watched by more than 20 million people in France [1], and the Germany vs. France semifinal was watched by almost 30 million people in Germany [1]. How does Hungary perform in comparison?

According to the MTVA (Media Services and Support Trust Fund), which operates the television channel M4 Sport, the first Hungarian match was watched by about 1.734 million people, the second by about 1.976 million, and the third group match by about 2.318 million people (according to the Hungarian Central Statistical Office (KSH), the population of Hungary was about 9.83 million in 2016 [2]). With these ratings, M4 Sport became the most watched television channel in Hungary during those days [3]. The performance of the Hungarian national football team was beyond expectations and raised interest, even among those who generally do not follow football matches. In this paper, we aim to answer the question of whether it is possible to measure/correlate this interest using a mobile phone network?

Mobile phones can function as sensors that detect the whereabouts and movements of their carrier. In this day and age, practically everyone has a mobile phone, which makes it possible to use large-scale analyses. With enough data, general mobility customs and reactions to events can also be studied. The first step is to prepare the data and select the

appropriate individuals for use in the study. Filtering the subscribers of CDR data sets is always a crucial step, and not just to eliminate inactive users: a subscriber who only appears a few times in a data set cannot be used for mobility analysis, but abnormally active subscribers can also bias the results. In particular, this is a problem if their location does not change, as CDR data may not only contain records for cell phones that are carried by people.

Csáji et al. took into account subscribers who had at least 10 sessions of activity during the observation period (15 months) [4]. Xu et al. chose to use subscribers who had at least one activity record for at least half of the days during the observation period [5]. Pappalardo et al. discarded subscribers who had only one location and used individuals who had at least half as many calls as there were hours in the data set. Furthermore, users with abnormally active (more than 300 calls per day) SIM cards were excluded [6]. In [7], we selected the SIM cards, which have activity on at least 20 days (out of 30). The daily mean activity number is at least 40 on workdays and at least 20 on weekends, but not more than 1000. The upper limit is especially important for removing SIM cards that possibly operate in mobile broadband modems, for example. Filtering by activity is not necessarily sufficient to make sure that only individuals are featured in the data set. Type Allocation Codes (TAC) can determine the type of device used and the exact model of cell phone used.

After the right subscribers have been selected, it is common practice to determine users' home and work locations [8–10]; then, between these two crucial locations, the commuting trends can be identified. Commuting is studied between cities [9,11–13] or within a city [7,14–20].

Apart from commuting and connectivity analyses, CDR processing is often used [21–28] for the detection of large social events. When thousands of people are in the same place at the same time, they generate a significant 'anomaly' in the data, whereas small groups usually do not stand out from the 'noise'. This is especially true when passive, transparent communication between the mobile phone device and the cell is not included in the data and only active communication (voice calls, text messages, and data transfer) are recorded.

In [26,27], mass protests are analyzed via mobile phone network data. In [21–23] and [28], the authors examined the location of stadiums where football matches had taken place. Traag et al. [21] and Hiir et al. [28] found that the mobile phone activity of attendees decreased significantly. In [21], z-score was used to express the deviation of activity during a social event from the average level. Xavier et al. compared the reported number of attendees of these events with the number detected. Furletti et al. also analyzed sociopolitical events, football matches, and concerts in Rome [24]. This paper focuses on football matches that took place in a remote country (France) and studies the fans' activity in Budapest.

Mobility indicators, such as Radius of Gyration or Entropy, are often calculated [5,6] to describe and classify subscribers' mobility customs. Furthermore, using mobility to infer about Social Economic Status (SES) is a major current direction of mobility analysis [5,7,13,29]. Cottineau et al. [29] explored the relationship between mobile phone data and traditional socioeconomic information gathered from the national census in French cities. Barbosa et al. found significant differences in the average travel distance between low- and high-income groups in Brazil [13]. Xu et al. [5] found opposite travel tendencies regarding the mobility of Singapore and Boston. In our previous work [7], we showed that the real estate price of home and work locations characterize mobility and validated our results using census data. In this paper, the price and age of subscribers' mobile phones are proposed to be used as socioeconomic indicators. While Blumenstock et al. used call history as a factor in socioeconomic status [30], Sultan et al. [31] applied mobile phone price as a socioeconomic indicator and identified areas where more expensive phones appear more often; however, only manually collected market prices were used.

Mobile phone network data have also been used to analyze human mobility during COVID-19 pandemic and the effectiveness of restrictions imposed. Willberg et al. identified a significant decrease in the population presence in the largest cities of Finland after the

lockdown compared to a usual week [32]. Bushman et al. analyzed compliance with social distancing policies in the US using mobile phone data [33]. Gao et al. found a negative correlation between stay-at-home distancing and the rate of increase in COVID-19 infections [34]. However, these analyses still might not be common enough. Oliver et al. asked: 'Why is the use of mobile phone data not widespread, or a standard, in tackling epidemics?' [35]. This matter, however, is not within the scope of this paper.

In this study, we analyzed mobile phone network activity before, during, and after matches played by the Hungarian national football team. The Call Detail Records (CDR), which were analyzed in this study, were recorded in Budapest; however, the matches took place in France. We present another example of social sensing using CDRs in both an indirect and a direct way. For the indirect method, we studied the mobile phone activity of sport fans residing in Budapest during matches played in France. For the direct method, we analyzed the spontaneous festival that took place on the streets of Budapest after the third match and the welcome event at the Heroes' Square from a data perspective.

The Call Detail Records were filtered by the Type Allocation Codes (TAC) to remove Subscriber Identity Module (SIM) cards that do not operate in mobile phones and thus that are not used by actual people. The price and age of the cell phones were also analyzed in relation to the subscribers' ages and mobility customs.

The contributions of this paper are briefly summarized as follows:

1. Fusing CDR data sets with mobile phone prices and release dates.
2. Filtering out SIM cards that do not operate in mobile phones.
3. Demonstrating connections between phone prices and mobility customs.
4. Proposing the use of mobile phone price as an SES indicator.
5. Attendees of the large social events were compared to the rest of the subscribers based on their mobility and SES.

The rest of this paper is organized as follows. The utilized data are described in Section 2; then, in Section 3, the applied methodology is summarized and in Section 4 the results of this study are introduced. Finally, in Section 5, the findings of the paper are summarized and a conclusion is drawn.

## 2. Materials

Vodafone Hungary, one of the three mobile phone operators providing services in Hungary, provided anonymized CDR data for this study. The observation area was Budapest, the capital of Hungary and its agglomeration, and the observation period was one month (June 2016). In 2016 Q2, the nationwide market share of Vodafone Hungary was 25.3% [36]. This data set contains 2,291,246,932 records from 2,063,005 unique SIM cards and does not specify the type of activity taking place (voice calls, text messages, or data transfers).

Figure 1 shows the activity distribution between the activity categories of the SIM cards. The dominance of the last category, SIM cards with more than 1000 activity records, is even more significant. It can be seen that almost 27% of the SIM cards produce more than 91% of the total activity.

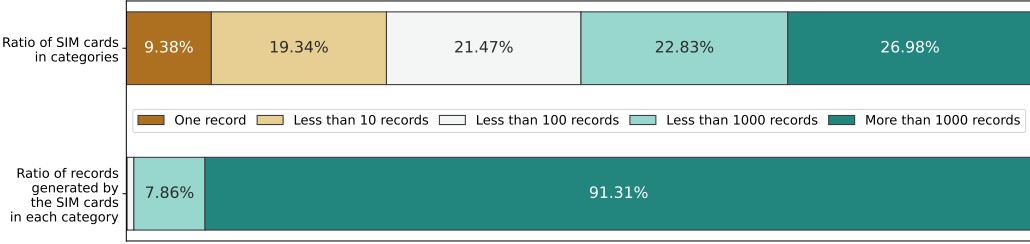

**Figure 1.** Subscriber Identity Module (SIM) cards categorized by the number of activity records. The SIM cards with more than 1000 activity records (26.98% of the SIM cards) provide the majority (91.31%) of the activity.

Figure 2 shows the SIM card distribution according to the number of active days. Only 34.59% of the SIM cards had activity on at least 21 different days. In total, 241,824 SIM cards (11.72%) appeared on at least two days, but the difference between the first and the last activity was not more than seven days. This may indicate the presence of tourists. High levels of tourism are usual during this part of the year.

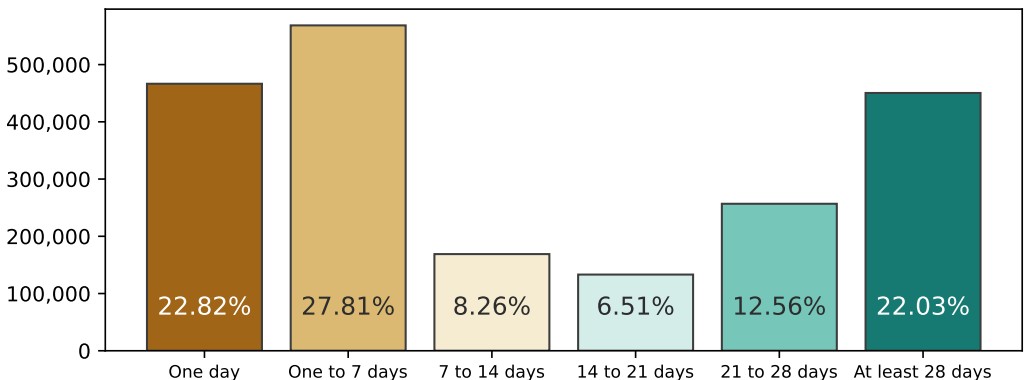

**Figure 2.** SIM card distribution by the number of active days.

The obtained data were in a 'wide' format and they contained an SIM ID, a timestamp, a cell ID, the base station (site) coordinates in WGS 84 projection, the subscriber (age, sex) and subscription details (consumer/business and prepaid/postpaid), and the Type Allocation Code (TAC) of the device. The TAC is the first 8 digits of the International Mobile Equipment Identity (IMEI) number. This is allocated by the GSM Association and uniquely identifies the mobile phone model.

Type Allocation Codes are provided for every record because a subscriber can change their device at any time. Naturally, most of the subscribers (95.71%) used only one device during the whole observation period, but there were some subscribers—maybe mobile phone repair shops—who used multiple devices (see Figure 3a). As part of the data cleaning process, the wide format was normalized. The CDR table contained only the SIM ID, the timestamp, and the cell ID. A table was formed from the subscriber and subscription details and another table was created to track the device changes of the subscriber.

While the subscription details were available for every SIM card, the subscriber information was missing in slightly more than 40% of the cases, presumably because of the subscribers' preferences regarding the use of their personal data. Figure 3b shows the age distribution of the subscribers for whom data regarding their subscription type were available (58.65%). Note that this may not represent the true age distribution of the population or even the distribution of the customers of Vodafone Hungary, as users are allowed to have multiple subscriptions and the actual user of the phone may differ from the owner of the subscription. Nevertheless, it was clear that among elderly people, prepaid subscriptions were more popular.

Figure 4 shows the number of daily activity records obtained during the second half of the month. Weekends (brown bars) showed significantly less activity, with the activity during the matches being higher compared to the weekday or weekend activity average.

Although the data contained cell IDs, only the base station locations, where the cell antennas were located, were known. As a base station usually serve multiple cells, these cells were merged by the serving base stations. After the merge, 665 locations (sites) remained with known geographic locations. To estimate the area covered by these sites, Voronoi Tessellation was performed on the locations. This is a common practice [4,37–41] in CDR processing.

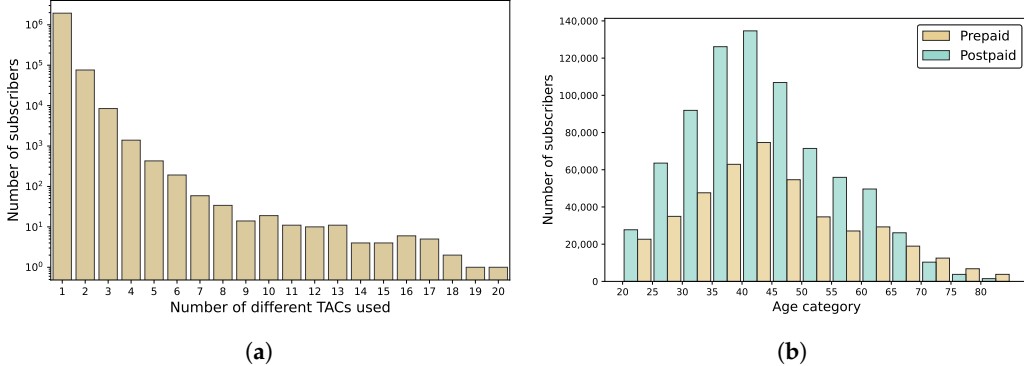

|     |     |
| :-: | :-: |
| (**a**) | (**b**) |

**Figure 3.** The number of different Type Allocation Codes (TAC) used by the subscribers (**a**), and the subscriber′ age distribution in respect of the subscription type (**b**).

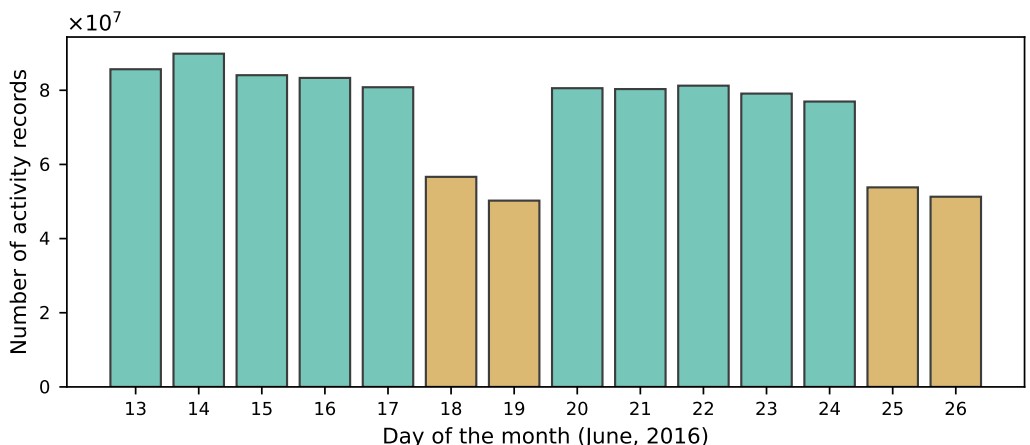

**Figure 4.** Number of daily activity records during two weeks of June 2016. The matches of the Hungarian national football team took place from 14 June to 26 June.

### 2.1. Resolving Type Allocation Codes

The socioeconomic status SES of the members in the celebrating crowd was intended to be determined according to the type of mobile device they used. The preliminary assumption was that the price of the mobile phone would represent the SES of a person.

To the best of our knowledge, there is no publicly available TAC database that can be used to resolve the TACs for the manufacturer and model, although some vendors (e.g., Apple, Nokia) publish the TACs of their products. The exact model of the phone must be known in order to determine how recent and expensive the mobile phone is. However, this is not sufficient to determine how much the cell phone actually cost the subscriber, as they could have bought it on sale or at a discount via their operator in exchange for signing an x-year contract. Still, the consumer price should designate the order of magnitude of the phone price.

The data set of TACs provided by '51Degrees' was used; this data set represents the model information in three columns: 'HardwareVendor', 'HardwareFamily', and 'HardwareModel'. The company mostly deals with smartphones that can browse the web, so feature phones and other GSM-capable devices are usually not covered by the data set. Release date and inflated price columns are also included, but these are usually not known, making the data unsuitable for use alone.

Although they cannot be separated by type, the CDR data contain not only call and text message records but also records of data transfers. Furthermore, some SIM cards do not operate in phones, but in other—often immobile—devices, such as 3G routers or modems. 51Degrees managed to annotate several TACs as modems or other non-phone devices. This was extended by a manual search of the most frequent TACs. We found

324,793 SIM cards that were used in only one device during the observation period and that operated in a non-phone device.

### 2.2. Fusing Databases

As a more extensive mobile phone price database, the GSMArena database [42] was used, that has a large and reliable database that has also been used in other studies [43,44]. The concatenation of the brand and model fields of the GSMArena database could serve as an identifier of the database fusion. 51Degrees stores the hardware vendor, family, and model of the mobile phone, where the hardware family often contains a marketing name (e.g., Apple, iPhone 7, A1778). As these fields are not always properly distinguished, the concatenation of the three fields may contain duplications (e.g., Microsoft, Nokia Lumia 820, Lumia 820). Thus, for the 51Degrees records, three identifiers were built using the concatenation of fields (i) vendor + family, (ii) vendor + model, and (iii) vendor + family + model. All three versions were matched against the GSMArena records.

Another step of the data cleaning was to correct the name changes. For example, BlackBerries were manufactured by RIM (e.g., RIM, BlackBerry Bold 9700, RCM71UW), but later the company name was changed to BlackBerry; the database records are not always consistent in this matter. The same situation occurred due to the acquisition of Nokia by Microsoft.

To match these composite identifiers, simple string equality cannot be used due to writing distinctions; thus, Fuzzy String match was applied using the FuzzyWuzzy Python package, which makes use of Levenshtein Distance to calculate the differences between strings. This method was applied for all the three identifiers from the 51Degrees data set and the duplicated matches (e.g., when the family and the model were the same) were removed. Mapping the GSMArena database to the 51Degrees added phone price and release date information to the TACs, which could be merged with the CDRs.

From the GSMArena data, two indicators were extracted: (i) the price of the phone (in EUR) and (ii) the relative age of the phone (in months). The phone price was left intact without taking into consideration the depreciation, and the relative age of the phone was calculated as the difference between the date of the CDR data set (June 2016) and the release date of the phone.

## 3. Methodology

A framework introduced in our earlier work [7] was applied to process the mobile phone network data. The CDRs were normalized and cleaned and the mobility metrics (Section 3.1) were determined for every subscriber. The records could be filtered spatially and temporally; both of these types of filtering were applied in this work. Additionally, a group of SIM cards could be selected from the activity records.

Only temporal filtering was applied to visualize the activity trends during the football matches. The activity of the subscribers is utilized from the whole observation area. The time series also illustrate the activity two hours before and after the matches. For the celebration after the Hungary vs. Portugal match, spatial and temporal filtering was applied to select the area of interest (Budapest downtown) in the given time interval.

To determine the activity levels for the sites, the match-day activity, the average weekday activity (without the match day), and the Z-scores were determined for the sites of the area of interest (downtown) in the selected time interval (20:15–20:20). The standard score (or Z-score) is defined as $z = \frac{x-\mu}{\sigma}$, where $\mu$ is the mean and $\sigma$ is the standard deviation. We observed that the standard deviation would be higher if the target day activity were not removed from the reference average; consequently, the Z-score would be lower and the relative differences would be less consistent. The histogram of the Z-score was generated for the selected sites (Figure 5) to determine the activity categories. A value of zero means that the activity level is equal to the average, but a wider interval (between −2 and 2) is considered to be average to allow some variation. Sites with a Z-score of between 2 and 8 are considered to have a high level of activity during the given time interval. There are

also sites with either low (below −2) or very high activity (over 8). The same method was applied for the site at Heroes' Square, but as the area of interest and the event differ, the thresholds are not the same (see Section 4.5).

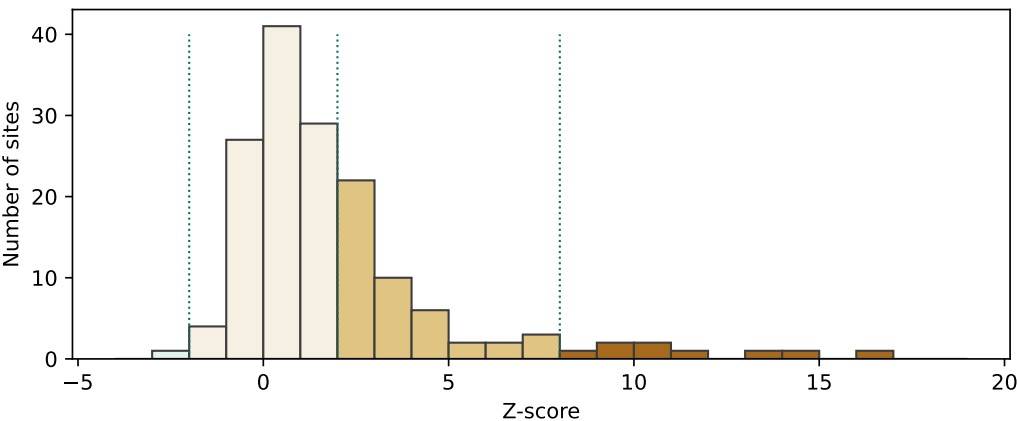

**Figure 5.** Z-score distribution of the downtown sites with the activity level thresholds of −2, 2, and 8, during the spontaneous festival.

The groups of football fans were formed from the subscribers based only on activity recorded during the Hungary vs. Portugal match. The owner of non-phone SIM cards that were active after at least two goals were considered to be active football fans. The properties of these subscribers—including the age, mobility metrics, phone age, and phone price—were compared to those of the rest of the subscribers.

### 3.1. Mobility Metrics

The metrics of Radius of Gyration and Entropy were used to characterize human mobility. These indicators were determined for every subscriber, omitting SIM cards that were operating in non-phone devices. In this study, locations were represented by the base stations.

The Radius of Gyration [45] is the radius of a circle in which an individual (represented by a SIM card) can usually be found. It was originally defined in Equation (1), where $L$ is the set of locations visited by the individual, $r_{cm}$ is the center of mass of these locations, and $n_i$ is the number of visits to the i-th location.

$$r_g = \sqrt{\frac{1}{N} \sum_{i \in L} n_i (r_i - r_{cm})^2} \tag{1}$$

The entropy characterizes the diversity of locations visited using an individual's movements, defined as Equation (2), where $L$ is the set of locations visited by an individual, $l$ represents a single location, $p(l)$ is the probability of an individual being active at a location $l$, and $N$ is the total number of activities taken part in by an individual [29,37].

$$e = -\frac{\sum_{l \in L} p(l) \log p}{\log N} \tag{2}$$

### 3.2. Socioeconomic Status

In our earlier work [7], the real estate price of subscribers' home locations were used to describe their socioeconomic status. In this study, the CDRs were enriched by phone prices and the phone price was assumed to be able to be applied as a socioeconomic indicator. To demonstrate the applicability of the use of mobile phone price as a socioeconomic indicator, it was examined with respect to the mobility indicators by applying Principal Component Analysis (PCA).

The SIM cards were aggregated using the subscriber age categories (5-year steps between 20 and 80), the phone price categories (100 EUR steps to 700 EUR), and the Radius of Gyration and Entropy categories. For the Radius of Gyration, 0.5 km distance ranges were used between 0.5 and 20 km, while the Entropy values were divided into twelve bins between 0.05 and 1.00. The structure of the data used for the Principal Component Analysis can be defined as follows.

A table was generated where every row consisted of 40 columns representing 40 Radius of Gyration bins between 0.5 and 20 km and 20 columns representing 20 Entropy bins between 0.05 and 1.00. The subscribers belonging to each bin were counted and their cardinality was normalized by metrics in order to compare them. The categories were not explicitly labeled, so the subscriber age and phone price descriptor columns were not provided in the PCA algorithm. The same table was constructed using weekend/holiday metrics and its rows were appended after the weekday ones.

When the PCA was applied, the 60-dimension vector was reduced to two dimensions based on the mobility customs, where the bins were weighted by the number of subscribers. The cumulative variance of the two best components was about 61% (see Figure 6a). The bins representing the two new dimensions (PC1 and PC2) were plotted (see Figure 7) and the markers were colored by the phone price. Marker sizes indicate the subscriber age category and larger markers were used for younger subscribers.

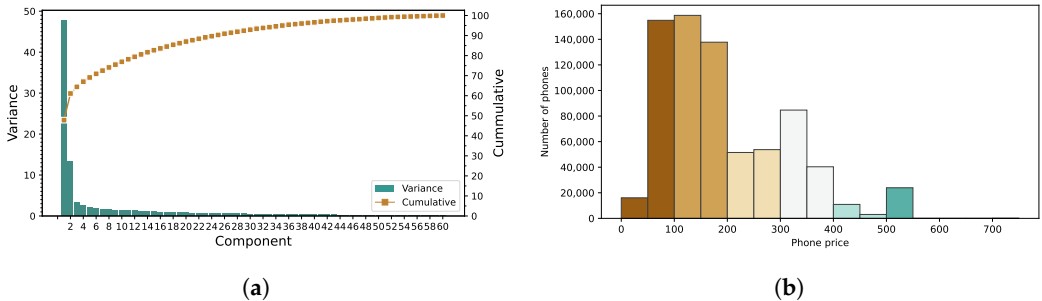

(**a**)                                    (**b**)

**Figure 6.** The Pareto histogram for the 60 components of the Principal Component Analysis (PCA) (**a**), and the phone price distribution (**b**).

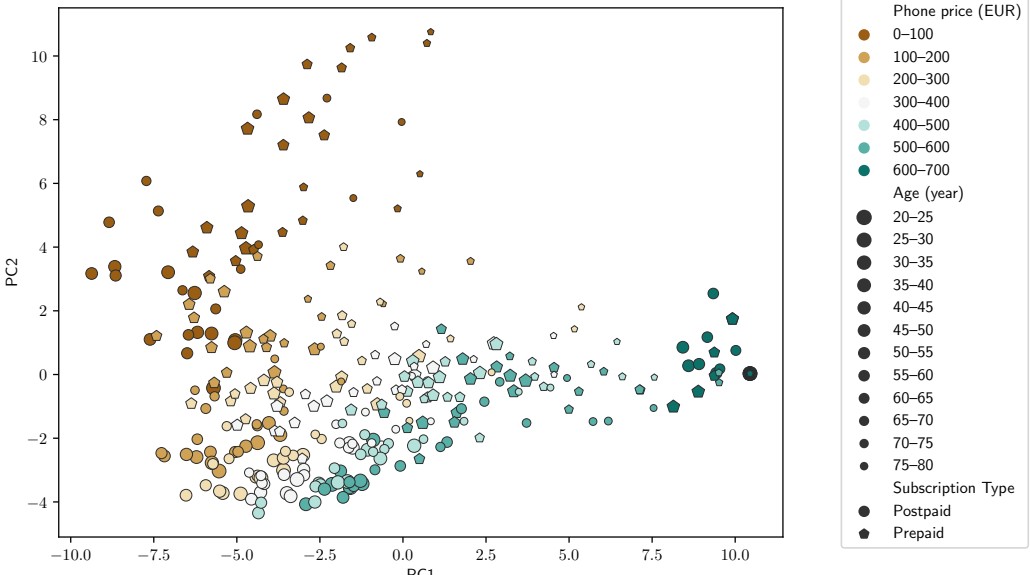

**Figure 7.** Scatter plot of the 2-component PCA. Marker sizes indicate subscriber age category, where the color represents the phone price category and the subscription type (prepaid/postpaid) is distinguished by the marker type.

## 4. Results and Discussion

As Figure 7 shows, the markers are clustered by color—in other words, the phone price, which is proportional to PC1 but inversely proportional to PC2. Within each phone price group, the younger subscribers (larger markers) are closer to the origin, indicating that the mobility custom of the younger subscribers differs from that of the elderly customers, although this difference is smaller within the higher price categories. This finding coincides with [46], where Fernando et al. found a correlation between subscribers' age and mobility metrics.

To give context to Figure 7, Figure 6b shows the phone price distribution, where most of the phones are within the 50–200 EUR range. Note that there are only a few phones over 550 EUR, but the owners of those have significantly different mobility patterns.

Figure 7 not only shows that the phone prices form clusters, but also reveals the effect of the subscription type on the mobility. Within the phone price categories, except for the highest with only a very few subscribers, the postpaid groups are usually closer to the origin. Prepaid subscriptions are usually bought by those who do not use their mobile phone extensively, and it seems that people with a prepaid subscription have similar mobility customs to people with less expensive phones but a postpaid subscription. This is most notable at $(-6, 2)$ and $(-5, -1)$.

Sultan et al. identified areas in Jhelum, Pakistan, where the more expensive phones appear more often [31]. Using the same method, Budapest and its agglomeration were evaluated and the average phone prices were determined for every site based on the activity records. The ground truth was that the real estate prices were higher at the Buda side (west of the river Danube) of Budapest and downtown [7], and this tendency can be clearly seen in Figure 8. The airport area had a significantly higher average than its surroundings, which is not surprising. The spatial tendencies of the mobile phone price, along with the result of the PCA (Figure 7), clearly demonstrate the expressiveness of the use of phone price as a socioeconomic indicator.

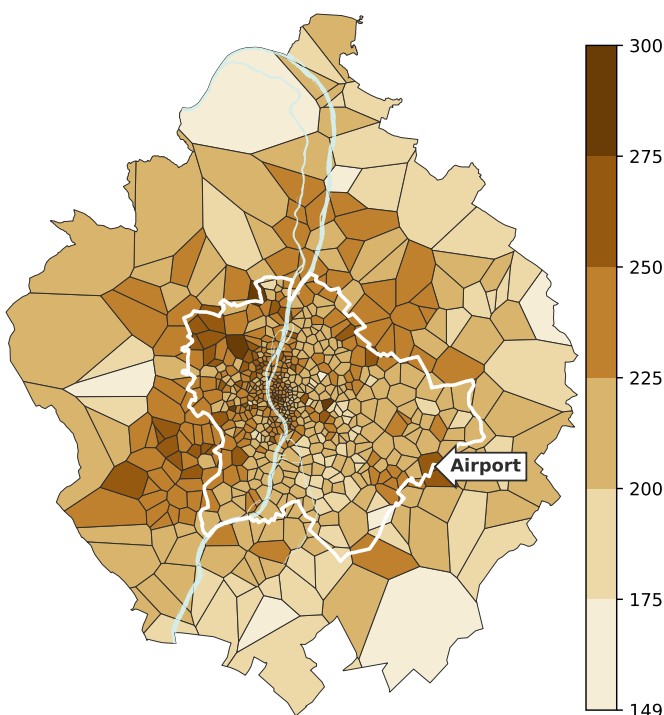

**Figure 8.** Average price (in EUR) of the mobile phones that generated the activity records in each site during the whole observation period (June 2016).

The rest of this section examines the results in the time order of the Hungarian Euro 2016 matches.

### 4.1. Austria vs. Hungary

The first match against Austria (Figure 9) started at 18:00 on Tuesday, 14 June 2016. Before the match, the activity level was significantly higher than the average of the weekdays and later decreased until half-time. During the second half, the activity level dropped to the average, which indicated that more people started to follow the match. Right after the Hungarian goals, two significant peaks in the activity were observed, which indicates increased attention and the massive usage of mobile devices during the match.

As the data source could not distinguish the mobile phone activities by type, it was not possible to examine what kind of activities caused the peaks. It was supposed that the activity was mostly data transfers or text messages and not phone calls. We thought that it was unlikely that users would call someone during the match just because of a goal, but sending a line of text via a popular instant messaging service was very feasible.

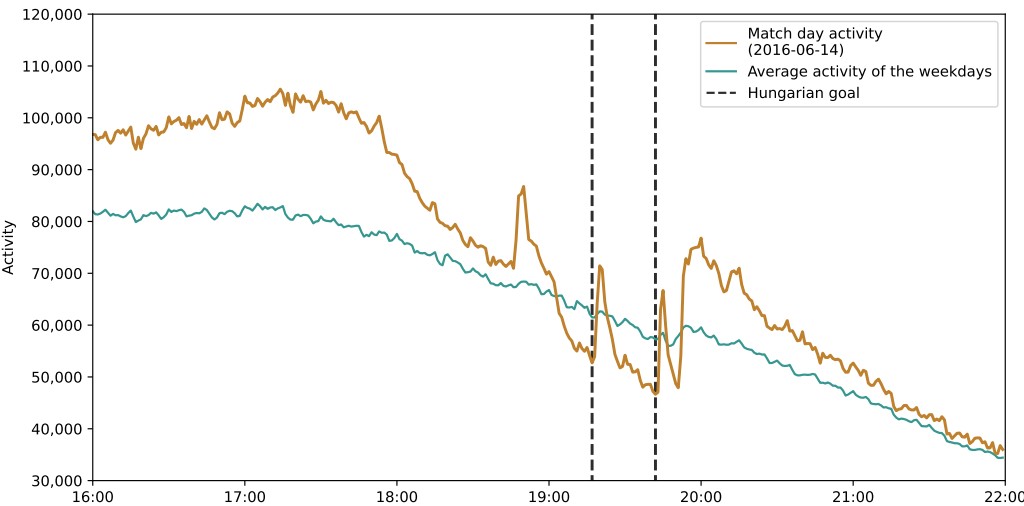

**Figure 9.** Mobile phone activity during and after the Austria–Hungary Euro 2016 match in comparison with the average activity on weekdays.

### 4.2. Iceland vs. Hungary

The match against Iceland was played on Saturday, 18 June 2016. Figure 10 shows the mobile phone activity levels before, during, and after the match. As the weekend activity is generally lower (see Figure 4), the average of the weekdays was used as a reference. The match began at 18:00, and from that point the activity level was significantly below average, except at the half-time break and, again, the peak after the Hungarian goal. Interestingly, the Icelandic goal did not result in such a significant peak; only a very moderate one can be seen in the time series.

Traag et al. [21] also found an activity drop during a game, but in that case the area of the stadium where the match was played was analyzed and no peak was found during the match.

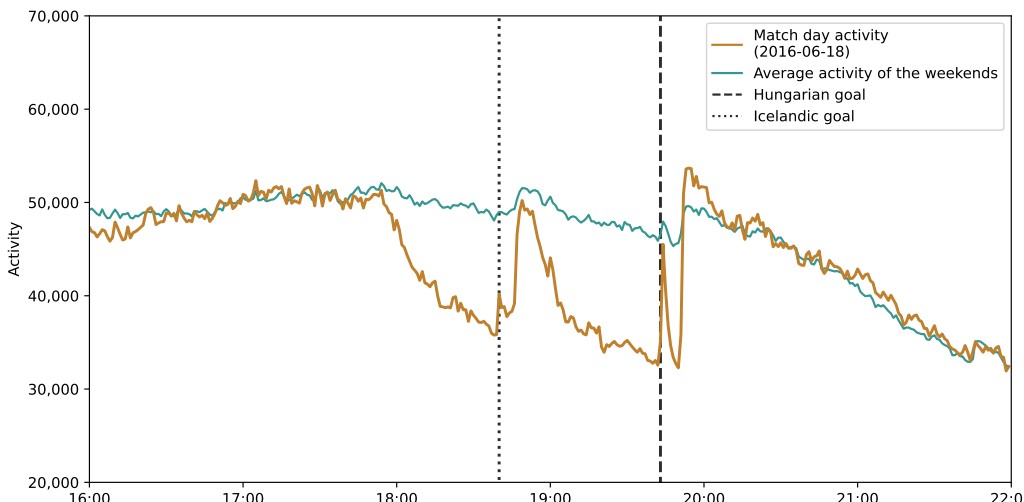

**Figure 10.** Mobile phone activity during and after the Iceland–Hungary Euro 2016 match in comparison with the average activity on weekends.

### 4.3. Hungary vs. Portugal

On Wednesday, 22 June 2016, as the third match of the group state of the 2016 UEFA European Football Championship, Hungary drew with Portugal. Both teams scored three goals, and with this result Hungary won their group and qualified for the knockout phase. During the match, the mobile phone activity dropped below the average, but the goals against Portugal resulted in significant peaks, especially the first one (see Figure 11). On the other hand, the Portuguese equalizer goal(s) did not cause significant changes in the activity. In the second half, the teams scored four goals in a relatively short time period, but only the Hungarian goals resulted in peaks. This observation suggests that the football fans had a notable influence on the mobile network traffic.

After the match, the activity level was over the average, which might represent the spontaneous festival that occurred in downtown Budapest. According to the MTI (Hungarian news agency), thousands of people celebrated in the streets starting from the fan zones, mainly Erzsébet square (Figure 12a), Margaret Island (Figure 12b), and Erzsébet square (Figure 12c) in the direction of Budapest Nyugati railway station. The Grand Boulevard was completely occupied by the celebrating crowd and public transportation was disrupted along those affected lines.

This social event was comparable to mass protests from a mobile phone network perspective. In an earlier work [47], we analyzed the mobile phone activity along the route of a mass protest. The activity of the cells was significantly higher when the protesters passed through the cell. In this case, however, the affected area was smaller and the sites along the Grand Boulevard were very busy at the same time after the game.

The activities of the sites (multiple cells aggregated by the base stations) in Budapest downtown are illustrated in Figure 13. The highlighted site covers Szabadság square (for the location, see Figure 12a), where one of the main fan zones was set up with a big screen. The activity curve actually followed the trends of the whole data set (see Figure 11). There was high activity before the match; during half-time; and, for a short period, after the match. During the match, the activity decreased except for four not so significant peaks around the goals.

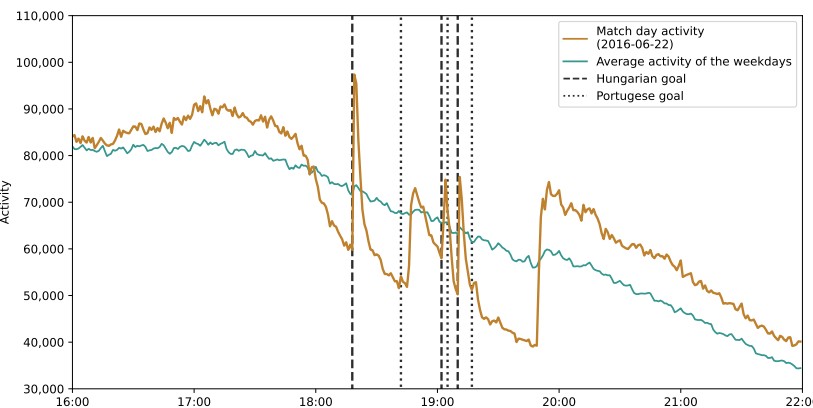

**Figure 11.** Mobile phone activity during and after the Hungary–Portugal Euro 2016 match in comparison with the average activity on weekdays.

In the highlighted site in Figure 13, almost 7 thousand SIM cards were detected between 17:00 and 20:00. The data showed that 53.57% of the subscribers were between 20 and 50 years old, while no age data were available for 33.49%.

After the match, there was a significant increase in activity in some other sites. These sites were (mostly) around the Grand Boulevard, where the fans marched and celebrated the advancement of the national football team to the knockout phase.

Figure 12 shows the spatial distribution of this social event using Voronoi polygons generated around the base stations locations. The polygons were colored according to the mobile phone network activity increase at 20:15 compared to average weekday activity. For the comparison, the standard score was determined for every base station within a 5-min temporal aggregation. The darker colors indicate the higher activity surplus in an area. The figure also shows the three main fan zones in the area, the routes of the fans (marked by arrows), and the affected streets.

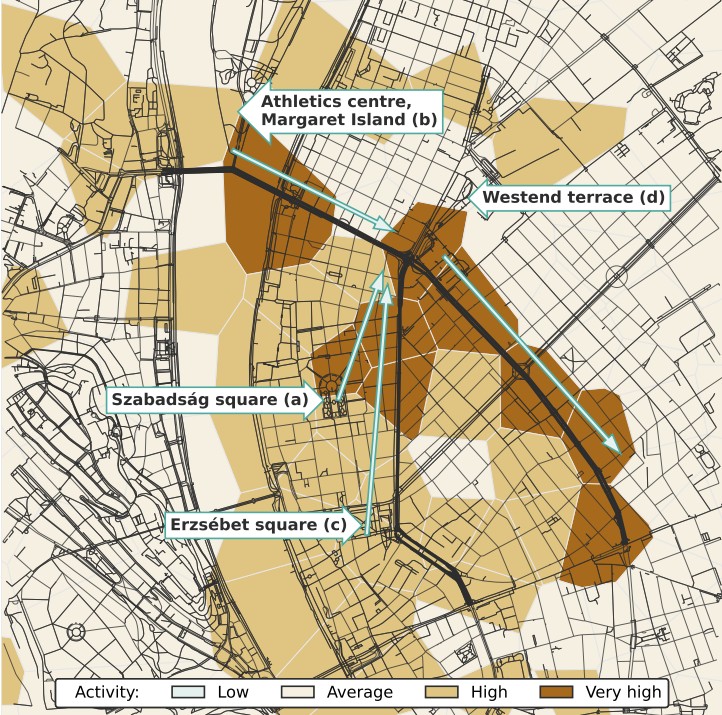

**Figure 12.** After the Hungary vs. Portugal football match, the fans, delirious with joy, filled the streets. The arrows show their route from the main fan zones to and along the Grand Boulevard. Voronoi polygons colored by the mobile phone network activity at the peak of the event at 20:15.

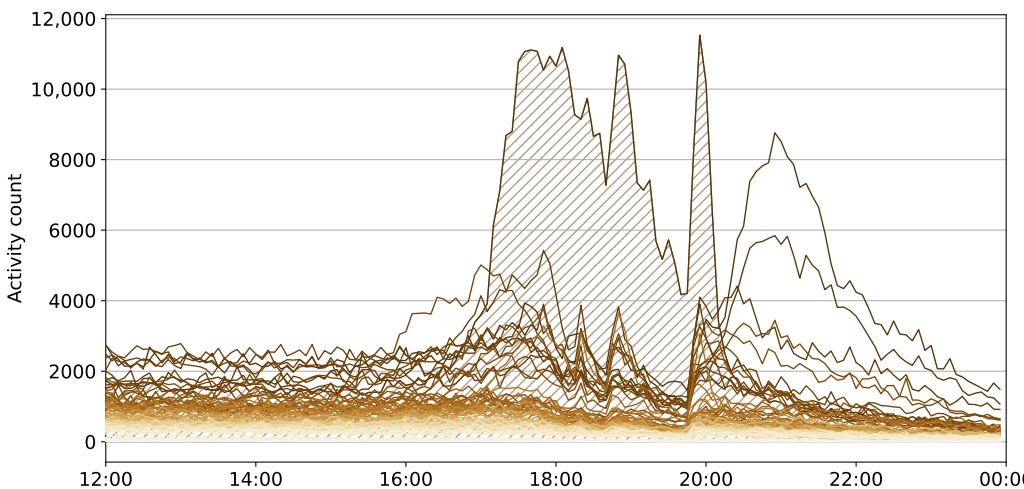

**Figure 13.** Site activities in Budapest downtown on the day of the Hungary vs. Portugal football match (22 June 2016). The highlighted site covers the Szabadság Square, where one of the main fan zones was set up.

Who Are Responsible for the Peaks?

Three Hungarian goals were scored during the match; hence, there were three peaks starting at 18:18, 19:02 and 19:18. All of them had about 5-min fall times. To analyze them, the SIM cards that were active during any two of the peaks were selected. Selecting SIM cards that were active during any of the peaks would also include many subscribers who could not be considered to be football fans. The participation of all the three peaks, on the other hand, would be too restrictive.

Figure 14a presents the activity of the selected 44,646 SIM cards and the owners of these cards, which may belong to the football fans. Removing these SIM cards from the data set should result an activity curve without peaks that is, at the same time, similar in tendency to the average activity. However, as Figure 14b shows, the activity still dropped during the match. Therefore, the 'football fan' category should be divided into 'active' and 'passive' fans from the mobile phone network perspective. Active fans are assumed to express their joy using the mobile phone network (presumably to access social media), causing the peaks. It seems that the passive fans ceased their other activities and watched the game, which caused some lack of activity compared to the average. By removing the active fans from the observed set of SIM cards, the activity level decreased in general (Figure 14b). However, this is not surprising; as these people reacted to the goals, they must often use their mobile phone network. There were also some negative peaks, indicating that the selection was not perfect.

Is there any difference between the active fans regarding their phone age and price compared to the other subscribers? Figure 15a shows the relative age of the phones with respect to the subscribers' behavior after the goals. No significant difference was found between the active fans and other subscribers, the median of the phone relative age was about two years, and there were some much older (nearly ten years old) phones in use. It should be noted that the older devices were used by elderly people. The price of the phones showed an opposite tendency: the younger subscribers owned more expensive phones (Figure 15b).

Naturally, not all of these 169,089 SIM cards (excluding the ones operating non-phone devices) generated activity after all the goals. A total of 83,352 devices were active after the first goal, 70,603 were active after the second, and 68,882 were active after the third. After at least two goals 44,646 and after all the three goals, only 9102 devices showed activity within 5 minutes.

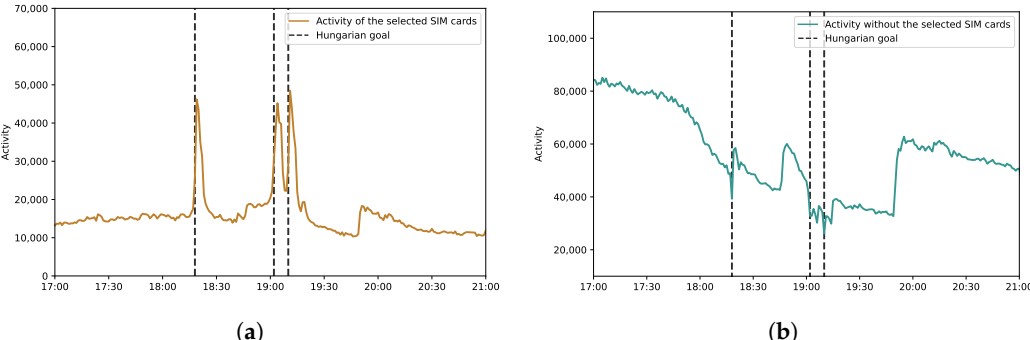

(a)

(b)

**Figure 14.** Mobile phone network activity of the SIM cards (fans) who showed activity right after any two of the Hungarian goals (**a**) and the activity of the other SIM cards, without the fans (**b**).

Why would these fans use their mobile phone network to access social media? If they were at home, they would have used a wired connection via Wi-Fi for mobile devices. In Hungary, 79.2% of the households had wired internet connection, according to the KSH [48], and this figure could be even higher in Budapest. However, if they were in fan zones—for example, in Szabadság Square—their use of the mobile network was more obvious.

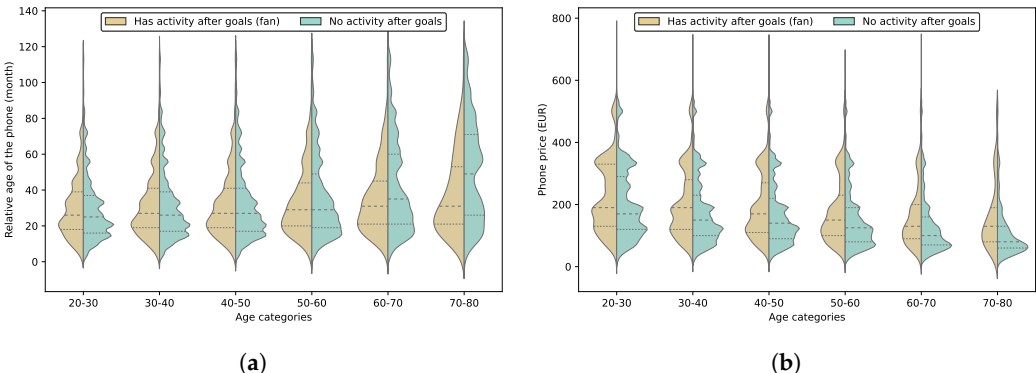

(a)

(b)

**Figure 15.** Mobile phone relative age (**a**) and the price distributions (**b**) in different age categories, comparing the fans who had activity right after any two of the Hungarian goals and the rest of the SIM cards.

As Figure 15 shows, there was no significant difference in the phone age between the active football fans and the rest of the subscribers. The medians were almost the same within the young adult and the middle-age categories, but elders tended to use older devices, especially those who did not react to the goals. The active football fans' median phone price was 180 EUR, in contrast to the 160 EUR median of the rest of the subscribers. However, the older subscribers tended to use less expensive phones. This tendency was also present within the football fans, but stronger within the other group.

Figure 16 illustrates the mobility metrics in different age categories and also compares the football fans and the rest of the subscribers. The Radius of Gyration median was almost the same in all the age categories and groups. The Entropy medians showed a notable difference between the two groups but did not really change between the age categories. This means that the mobility customs of the football fans who used the mobile phone network more actively were similar, regardless of the subscribers' age.

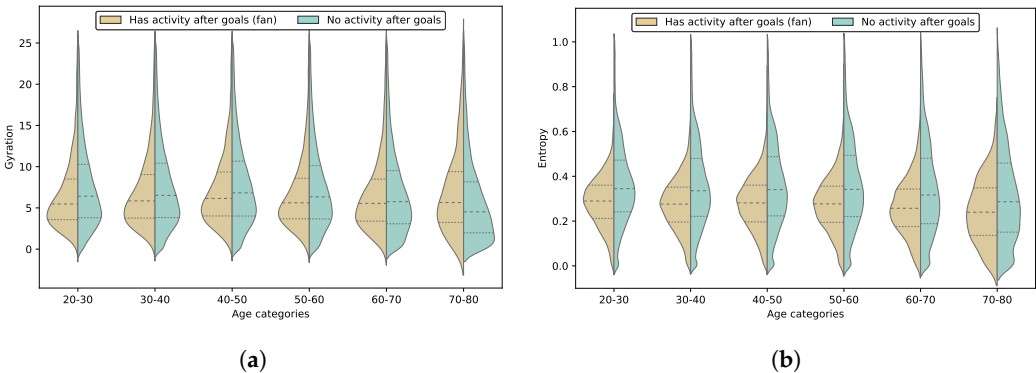

(**a**)           (**b**)

**Figure 16.** Radius of Gyration (**a**) and Entropy (**b**) distributions in different age categories, comparing the fans who had activity right after any two of the Hungarian goals and the rest of the SIM cards.

*4.4. Hungary vs. Belgium*

On Sunday, June 26, 2016, Hungary played the fourth and last Euro 2016 match against Belgium. Figure 17 shows the mobile phone network activity before, during, and after the match. During the match, the activity level was below the weekend average. The activity after the match was slightly higher than average, since the match ended late on Sunday, when the activity average is usually very low. This activity surplus may only indicate that the fans were simply leaving the fan zones and going home.

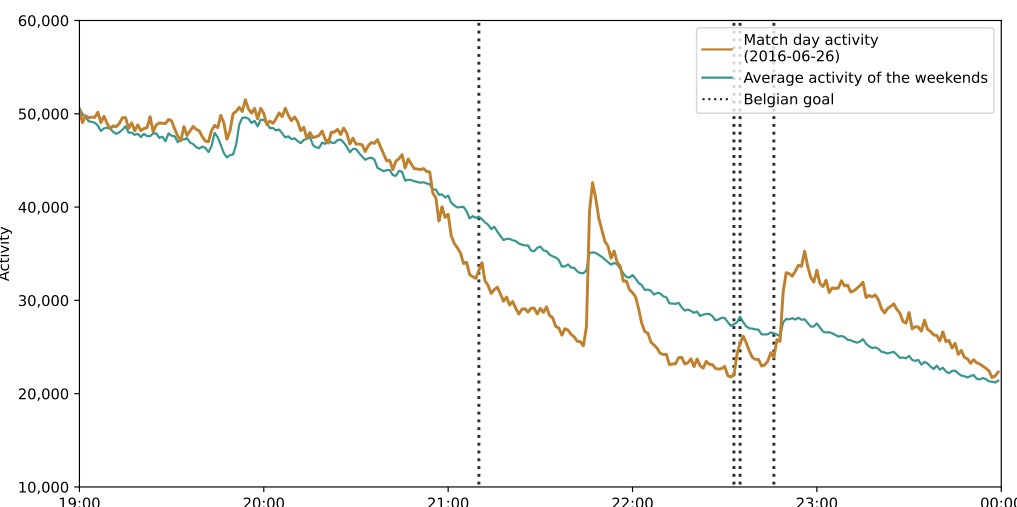

**Figure 17.** Mobile phone activity during and after the Hungary–Belgium Euro 2016 match in comparison with the average activity of the weekends.

*4.5. Homecoming*

The Hungarian national football team returned to Budapest on 27 June 2016. A welcome event at the Heroes' Square was held, where the football fans could greet the national football team. According to the M4 Sport television channel, approximately 20 thousand people attended the event [49]. Between 18:00 and 19:30, there were 4246 unique, non-phone SIM cards active in the site covering the Heroes' Square. In total, 3425 were known to use smartphones based on the operating system column of the GSMArena data set.

The cells of this base station covered a larger area, so not all of these subscribers actually attended the event, but on the other hand it was not compulsory to use mobile phones during this event. Supposing that the mobile phone operator preferences among the attendees corresponded to the nationwide trends in 2016, there could even have been about 17 thousand people, as the data provider had a 25.3% market share [36].

Figure 18b shows a part of District 6; the city park with the Heroes' Square and the Voronoi polygons of the area were colored according to the Z-score values to indicate the

mobile phone activity in the area at 18:35. The activity was considered low below −1, average between −1 and 1, high between 1 and 2.5, and very high above 2.5. Figure 18a shows the mobile phone network activity (upper) and the Z-score (bottom) of the site covering Heroes' Square. It is clear that during the event, the activity was significantly higher than the weekday average, and the Z-score values also followed that trend.

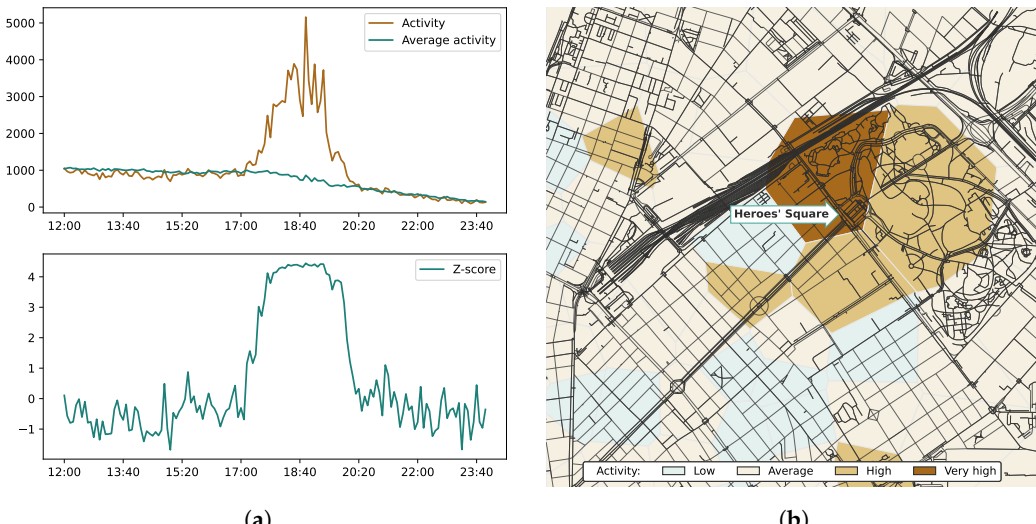

(**a**)        (**b**)

**Figure 18.** Mobile phone network activity at Heroes' Square and the surrounding neighborhood during the welcoming event for the Hungarian national football team. The activity and the Z-score of the site at Heroes' square (**a**), and the spatial view at 18:35 (**b**).

*4.6. Limitations*

We associated subscribers' SES with the release price of their cell phones; however, they did not necessarily buy their phones at that price. Many people buy their phones on sale or at a discounted price via the operator in exchange for signing an x-year contract.

Additionally, subscribers can change their phone devices at any time. We took into consideration only those subscribers who had used only one device during the observation period or who had a dominant device that generated most of the activity records of the given subscriber.

We fused three data sets to exclude non-phone SIM cards, but the list of identified devices is not complete. There remained devices for which models are unknown, and there were also phones for which the release date and price were unknown. It was not possible to determine the SES of these subscribers using the proposed solution.

*4.7. Future Work*

Although the current solution, which involves selecting the SIM cards of football fans— in other words, the SIM cards that caused the peaks—gives a reasonable result, it could be improved by analyzing the activity during the whole observation period. For example, it could be improved by applying a machine learning technique.

Extending the list of the non-phone TACs could also help us to refine our results, and combining the mobile phone prices with the real estate prices of the home location would most certainly enhance the socioeconomic characterization.

The relative age of the cell phone could be used as a weight for the phone price when applied as an SES indicator to distinguish between the phone price categories; an expensive but older phone is not worth as much as a newer one with the same price.

**5. Conclusions**

In this study, we demonstrated that mobile phone network activity precisely shadows football fans' behavior, even if the matches are played in another country. This analysis focused on people who followed the matches on TV (at home) or on big screens in fan

zones, but not in the stadium where the matches were actually played. The mobile phone network data and the mobile phone specification database were applied to characterize the SES of the football fans. The data fusion allowed us to remove a considerable number of SIM cards that operated in devices other than mobile phones from the examination. Although there were some still unidentified TACs in the data set, in this way we made sure that the activity records used in this study had a significantly higher possibility of being used by an actual person during the events.

The time series of mobile network traffic clearly show that the activity was below the average during the matches, indicating that many people followed their team. This observation is in line with the results of other studies [21–23,28], where the activity of the cell phones at the stadium where matches were played was analyzed. We also demonstrated that a remote football match can have notable effect on a mobile phone network. Moreover, the joy felt after the Hungarian goals was clearly manifested in the data as sudden activity peaks. Thus, we can state that CDR data are capable of being used in social-sensing applications.

The analysis of the spontaneous festival that took place after the Hungary vs. Portugal match and the welcoming event staged at the Heroes' Square are direct applications of social sensing and comparable to mass protests from a data perspective. During these events, the mobile phone network activity was significantly higher than the average level in affected areas.

The price of the mobile phone was proven to be an expressive socioeconomic indicator. It was capable not only of clustering the areas of a city, but also of distinguishing subscribers according to their mobility customs. On the other hand, it did not seem to affect their interest in football.

**Author Contributions:** Conceptualization, G.P.; methodology, G.P.; software, G.P.; validation, G.P.; formal analysis, G.P.; investigation, G.P.; resources, G.P. and I.F.; data curation, G.P.; writing—original draft preparation, G.P.; writing—review and editing, G.P. and I.F.; visualization, G.P.; supervision, I.F.; project administration, I.F.; funding acquisition, I.F. All authors have read and agreed to the published version of the manuscript.

**Funding:** This research supported by the project 2019-1.3.1-KK-2019-00007 and by the Eötvös Loránd Research Network Secretariat under grant agreement no. ELKH KÖ-40/2020.

**Institutional Review Board Statement:** Not applicable.

**Informed Consent Statement:** Not applicable.

**Data Availability Statement:** Not applicable.

**Acknowledgments:** The authors would like to thank Vodafone Hungary and 51Degrees for providing the Call Detail Records and the Type Allocation Code database used in this study. For plotting the map, OpenStreetMap data were used; these data are copyrighted by the OpenStreetMap contributors and licensed under the Open Data Commons Open Database License (ODbL).

**Conflicts of Interest:** The authors declare no conflict of interest. The funders had no role in the design of the study; in the collection, analysis, or interpretation of data; in the writing of the manuscript; or in the decision to publish the results.

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
