# Peer review of "Analyzing the Behavior and Financial Status of Soccer Fans from a Mobile Phone Network Perspective: Euro 2016, a Case Study"

_information, doi:10.3390/info12110468_

Round 1
Reviewer 1 Report
This work revolves around using call detail records and descriptive statistical methods to analyze soccer fans' behavior and financial status. The idea is interesting; however, the paper should be improved from different perspectives:
Abstract should be rewritten. It should include an introduction of the topic, limitation of previous studies, main objective of this research, and the implemented method.
Introduction needs to be rewritten; the authors don't clearly state the purpose and contribution of this paper. It should demonstrate the motivations, objectives, and advantages, and disadvantages of different scenarios. What insight or additional knowledge can this work provide so that researchers are not aware of?
Generally speaking, the paper is well-organized and well-written. However, some paragraphs do not read well.
The literature review should be enhanced by commenting such work as
- M. Ghahramani, M. Zhou and C. T. Hon, "Mobile Phone Data Analysis: A Spatial Exploration Toward Hotspot Detection," in IEEE Transactions on Automation Science and Engineering, vol. 16, no. 1, pp. 351-362, Jan. 2019.
- M. Ghahramani, M. Zhou and C. T. Hon, "Extracting Significant Mobile Phone Interaction Patterns Based on Community Structures," in IEEE Transactions on Intelligent Transportation Systems, vol. 20, no. 3, pp. 1031-1041, March 2019.
- X. Peng, L. Liu and L. Zhang, "A Hive-Based Retrieval Optimization Scheme for Long-Term Storage of Massive Call Detail Records," in IEEE Access, vol. 8, pp. 431-444, 2020.
The authors should provide more relevant references and clearly distinguish their novelty compared with other research work.
More interpretation is expected—for example, comparing the significance of changes in mobility metrics and the comparison of diversity quartiles.
The potential link between transportation diversity and intra-urban human mobility can be discussed.
In summary, I think the paper has not contributed from a theoretical perspective, but it has some publishable material. I encourage the authors to revise and resubmit the paper.
Author Response
Response to Reviewer 1 Comments
The abstract has been updated.
The introduction has been rewritten:
- the former "Introduction" and "Literature Review" sections has been merged
- it has been extended in three directions
1. commuting and connectivity, including two papers of the suggested three. Although this article does not focus on commuting, in contrast with our previous paper (https://www.mdpi.com/2220-9964/10/5/328), this is indeed an important application that should be mentioned. See lines 54-57.
2. Filtering SIM cards/subscribers are commented with more details, as we proposed an approach for this problem (lines 41-53)
3. socioeconomic indicators are also detailed, to compare our approach with the former ones (lines 74-88)
- it contains more then ten new citations.
In this paper we do not wish to detail our data processing framework, so the third suggested paper has not been included.
The contributions are now explicitly listed, lines 110-116.
The methodology is extended, hoping that the approach is now clearer.
Lines 221-231 and 282-286 are added or heavily modified, referencing the figures in which the result can be seen.
Two new figures has been added, the first (Figure 8) directly reflects to the literature (citation 31), to highlight the phone price as a socioeconomic indicator in a spatial context. Figure 6 is commented with more details 290-295.
Figure 15 has been also added to support our findings in respect of football fans' the mobility. Lines 428-433
A subsection of Limitations is added (lines 461-472) and the Future Work is extended (lines 481-483).
While discussing our findings, the literature is commented. For example line 294, 306, 310, 338, 360.
Several other minor corrections has been also made to improve the overall quality of the paper.
Reviewer 2 Report
The authors present the results of mobile phone users' activity in a very interesting form. From one side, they used this information for explaining the groups creation (such as football matches) based on entropy estimation. On the other side, the authors analyst the age of phones and their price too.
The material is well visualized too.
The literature review is actual and conclusion section is relevant to the paper task.
It would be good to the methodology which time series algorithm authors used in the research.
As future research, it will be interesting to group users by phone price and analyze each group separatelly.
Author Response
Response to Reviewer 2 Comments
The methodology is extended, hoping that the approach is now clearer.
Lines 221-231 and 282-286 are added or heavily modified, referencing the figures in which the result can be seen.
Two new figures has been added, the first (Figure 8) directly reflects to the literature (citation 31), to highlight the phone price as a socioeconomic indicator in a spatial context. Figure 6 is commented with more details 290-295.
Figure 15 has been also added to support our findings in respect of football fans' the mobility. Lines 428-433
Besides this, based on Reviewer 1's comment, the abstract, the introduction has been updated (lines 41-57, 74-88, 110-116), a subsection of Limitations is added (lines 461-472), and several other minor corrections has been also made to improve the overall quality of the paper.
Round 2
Reviewer 1 Report
Thanks for addressing my comments. This work has not contributed from a theoretical point of view but it has some publishable material.